# Primate homologs of mouse cortico-striatal circuits

**Joshua Henk Balsters[1,2†]\*, Valerio Zerbi[2†]\*, Jerome Sallet[3], Nicole Wenderoth[2], Rogier B Mars[4,5]**

[1]Department of Psychology, Royal Holloway University of London, Egham, United Kingdom; [2]Neural Control of Movement Laboratory, Department of Health Sciences and Technology, ETH Zurich, Switzerland; [3]Wellcome Centre for Integrative Neuroimaging, Department of Experimental Psychology, University of Oxford, Oxford, United Kingdom; [4]Wellcome Centre for Integrative Neuroimaging, Centre for Functional MRI of the Brain (FMRIB), Nuffield Department of Clinical Neurosciences, John Radcliffe Hospital, University of Oxford, Oxford, United Kingdom; [5]Donders Institute for Brain, Cognition and Behaviour, Radboud University Nijmegen, Nijmegen, Netherlands

**Abstract** With the increasing necessity of animal models in biomedical research, there is a vital need to harmonise findings across species by establishing similarities and differences in rodent and primate neuroanatomy. Using connectivity fingerprint matching, we compared cortico-striatal circuits across humans, non-human primates, and mice using resting-state fMRI data in all species. Our results suggest that the connectivity patterns for the nucleus accumbens and cortico-striatal motor circuits (posterior/lateral putamen) were conserved across species, making them reliable targets for cross-species comparisons. However, a large number of human and macaque striatal voxels were not matched to any mouse cortico-striatal circuit (mouse->human: 85% unassigned; mouse->macaque 69% unassigned; macaque->human; 31% unassigned). These unassigned voxels were localised to the caudate nucleus and anterior putamen, overlapping with executive function and social/language regions of the striatum and connected to prefrontal-projecting cerebellar lobules and anterior prefrontal cortex, forming circuits that seem to be unique for non-human primates and humans.

**\*For correspondence:**
Joshua.Balsters@rhul.ac.uk (JHB);
valerio.zerbi@hest.ethz.ch (VZ)

†These authors contributed equally to this work

**Competing interests:** The authors declare that no competing interests exist.

## Introduction

Animal models are currently providing crucial insights into neural structure, function, and disorders. According to *Dietrich et al. (2014)* mice are the most used mammalian species in scientific research. However, with this comes a growing necessity for translational, comparative neuroscience to harmonise these results with our understanding of structure, function, and disease in the human brain. To date, formal comparisons of brain organisation across species have largely focussed on humans and non-human primates, in spite of the steady increase in rodent models in neuroscience (*Carlén, 2017*; *Ellenbroek and Youn, 2016*). This is partly driven by the lack of research in the different species using the same methods, but also by a distinct lack of consensus in terminology between research in rodents and research in primates, which is prohibitive of clear translation of results (*Laubach et al., 2018*). Here, we address the growing need to formally identify common brain circuits between mice, macaques, and humans using the same technique to determine the scope and limits of mouse translational models.

One key issue that has hampered the comparison of rodent and primate brain organisation has been establishing suitable comparative measures (*Preuss, 1995*). For instance, the discussion of the putative existence of a rat homolog of human prefrontal cortex, different authors have proposed

and dismissed a single connection to mediodorsal thalamus (*Preuss, 1995*; *Rose and Woolsey, 1948*), the presence of a granular layer IV (*Preuss, 1995*; *Uylings et al., 2003*), or equivalence of function (*Dalley et al., 2004*; *Laubach et al., 2018*; *Narayanan et al., 2013*) as diagnostics. These issues highlight the difficulties in understanding homologies across such distantly related species. Importantly, even if homology of areas is established, the homologous region will be embedded into a different large-scale network in the two species, which has consequences for the interpretation of translational results. Moreover, establishing similarity between anatomy and function is only useful if these two levels of description can be related. Ideally, we would want to understand what exactly is similar and different in the anatomical organisation of the different species' brains and how this relates to their behavioural abilities.

One approach to comparative neuroscience that has been successful at establishing similarities and differences on a continuous scale is that of matching areas across species based on their so-called 'connectivity fingerprint'. The term connectivity fingerprint was introduced by *Passingham et al. (2002)* to suggest that brain regions could be partitioned into functionally distinct brain areas based upon their unique set of connections, which in turn constrain the function of an area. For example, *Passingham et al. (2002)* showed that macaque premotor areas F3 and F5 possess unique connectivity fingerprints, which correspond to their unique functional responses based on electrophysiology (neural firing during memory guided vs visually guided tasks respectively). Whilst connectivity fingerprints were previously used as a way to distinguish between functionally distinct regions within one brain, *Mars et al. (2016)* proposed that connectivity fingerprint matching could be used as a tool to identify similar brain areas across species. As case in point, *Neubert et al. (2014)* systematically compared connectivity fingerprints from multiple regions in the human ventro-lateral frontal cortex (vlFC) with connectivity fingerprints of regions defined in macaques. The results identified 11 vlFC brain regions with similar fingerprints in both species, and 1 brain region (lateral Frontal Pole; FPl) that was uniquely human. This approach has also been used to compare brain regions in the dorsolateral prefrontal cortex (PFC) (*Sallet et al., 2013*), medial PFC and orbitofrontal cortex (*Neubert et al., 2015*), parietal lobe (*Mars et al., 2011*), and temporoparietal cortex (*Mars et al., 2013*) in humans and non-human primates. However, to our knowledge, it has never been used to compare neuroanatomical connectivity patterns across humans, non-human primates, and rodents.

This approach is now feasible due to the availability of the same type of neuroimaging data from humans, macaques, and mice. Although tracer-based connectivity mapping is often considered to be the 'gold standard' for comparative neuroscience, these methods are too expensive and labour intensive to use in most species and for the entire brain (*Mars et al., 2014*). The substantial investment of both time and money often means that it is only possible to investigate a small number of subjects which reduces the power of statistical comparisons and the ability to make global inferences. Also, these invasive approaches are far less common in humans, making it difficult to harmonise findings across species using a common methodology. Resting state fMRI (rsfMRI) is increasingly employed as a non-invasive tool to measure connectivity in humans and non-human primates (*Neubert et al., 2015*; *Schaeffer et al., 2018*; *Vincent et al., 2007*) and the availability of mouse resting state data (*Grandjean et al., 2020*; *Zerbi et al., 2015*) invites an extension of this work to larger-scale between-species comparisons. The increasing application of rsfMRI across species is likely because of the high degree of consistency between connectivity profiles derived from tracers and connectivity derived using rsfMRI (*Grandjean et al., 2017*). In addition, rsfMRI repositories are making large numbers of datasets freely available making it possible to conduct analyses using a common methodology across species with samples sizes that are orders of magnitude larger than traditional tracer-based approaches (*Milham et al., 2018*).

Here, we will use connectivity fingerprint matching to compare cortico-striatal circuits in humans, macaques, and mice. The general architecture of cortical-striatal circuits, with partially separated loops connecting distinct parts of the striatum with distinct parts of the neocortex, seems well preserved across mammals (*Haber, 2016*; *Heilbronner et al., 2016*). However, the specific implementation can be expected to differ when the cortex has expanded in particular lineages, including the human (*Murray et al., 2016*). These circuits are particularly affected by a number of psychiatric conditions (*Bradshaw and Enticott, 2001*), some of which have been the target of recent mouse models (*Zerbi et al., 2018*). As such, cortico-striatal circuits are an ideal target to assess the feasibility of mouse-macaque-human translational studies.

## Results

### Establishing cortico-striatal connectivity fingerprints in mice

The connectivity fingerprinting approach requires extracting a rsfMRI timeseries from a seed region and comparing that with rsfMRI timeseries from a collection of target regions. The strength of the correlation between seed and target timeseries will be an indication of the strength of connectivity between those two regions, and the variability in connectivity across targets will produce the connectivity fingerprint. *Figure 1a* provides a schematic illustrating how mouse striatal seed regions were created using an independent tracer-based connectivity dataset from the Allen Institute (*Oh et al., 2014*). The strength of tracer connectivity from 68 cortical injection sites was established for each voxel in the mouse striatum and a hierarchical clustering algorithm was used to cluster voxels with similar and distinct connectivity patterns – an approach commonly referred to as connectivity-based parcellation (*Balsters et al., 2018*; *Balsters et al., 2016*; *Eickhoff et al., 2015*). *Figure 1b* shows the three clusters in the mouse striatum with unique connectivity patterns based on tracer data; Nucleus Accumbens (NAcc), medial caudoputamen (CP.m), and lateral caudoputamen (CP.l) and their functional connectivity maps.

As a complementary approach to establishing mouse striatal seeds, we transformed the tracer-based striatal parcellation of *Chon et al. (2019)* into our MRI reference space and created voxelwise functional connectivity maps for each of the 33 seeds. Using the same approach as above, we applied hierarchical clustering to assess the independence of the 33 functional connectivity maps. The 33 striatal seeds showed a high degree of overlap (i.e. similar functional connectivity patterns)

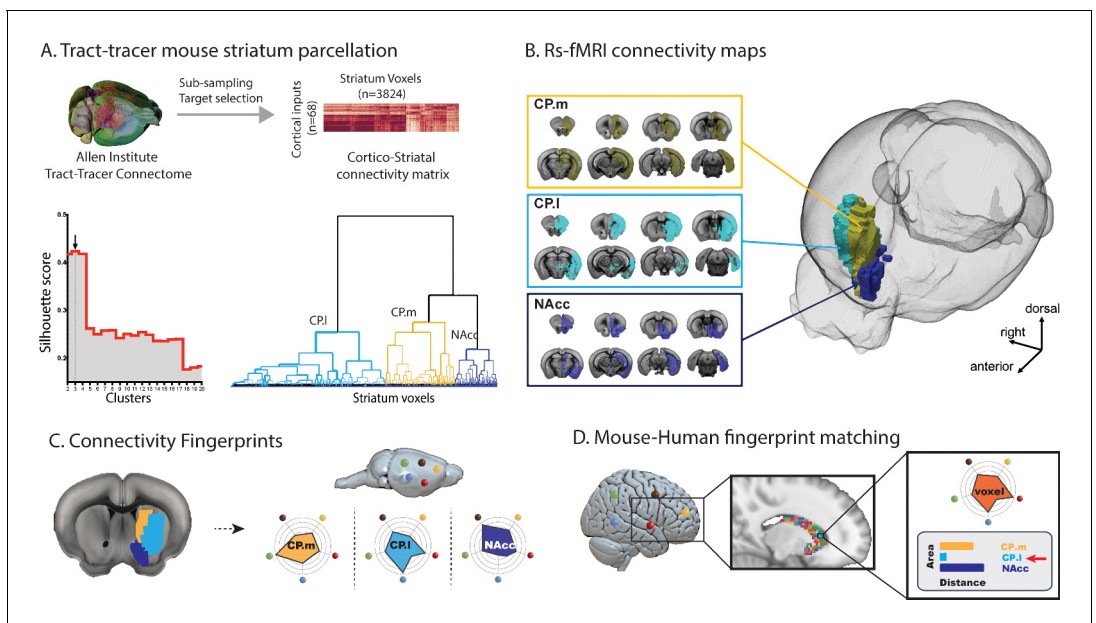

**Figure 1.** Images illustrating the creation of mouse striatal seeds, their connectivity, and connectivity fingerprint matching. (**A**) Schematic illustrating how mouse striatal seeds were created using tracer connectivity data from the Allen Institute. A connectivity matrix was extracted describing the volume of terminal label in each striatal voxel from 68 cortical injection sites. Correlating the values in this matrix established the similarity of connectivity fingerprints across striatal voxels. Hierarchical clustering was used to sort the correlation matrix into voxels with similar connectivity patterns (lower right dendrogram) and silhouette value was used to establish the number of clusters where within-cluster similarity and between-cluster differences were greatest (lower left plot). The arrow shows that the best solution according to silhouette value was a three cluster solution. (**B**) RsfMRI voxelwise connectivity maps for the three striatal seeds. (**C**) A schematic illustrating the process of extracted connectivity fingerprints for the three mouse striatal seeds. Connectivity fingerprints show the strength of connectivity (correlation of rsfMRI timeseries) between each striatal seed region and target regions outside the striatum **D**) In humans (or macaques), connectivity fingerprints were extracted from each voxel of the striatum - comparing the connectivity strength of striatal voxels with human homologs of the five target regions identified in mice. The similarity of each human voxel fingerprint can then be compared against each of the three mouse striatal fingerprints.

The online version of this article includes the following figure supplement(s) for figure 1:

**Figure supplement 1.** Figure showing the overlap between tracer-based and rsfMRI parcellations of the mouse striatum.

and were best characterized as four groups of seeds with distinct connectivity patterns (see *Figure 1—figure supplement 1*). Dice similarity showed that three of the four clusters were a match to the NAcc, CP.m, and CP.l previously defined using the voxelwise tracer data. One additional cluster was present in this approach within the CP tail; however, further analysis revealed that this cluster fell entirely outside of the striatal mask used in the previous analysis and included voxels in the bordering lateral amygdala nucleus. Given spatial smoothing involved in fMRI analysis, it is commonplace to remove voxels that border adjacent structures to avoid signal contamination which is known to have a significant effect on network detection (*Smith et al., 2011*). We suggest that this CP.tail segment is likely to be distinct because of signal contamination which is additionally problematic as the basolateral amygdala is one of our targets. Given the high consistency between both approaches, we will continue the analyses using the independent tracer-based seeds.

After creating mouse striatal seeds, we selected 12 target regions across cortical and subcortical regions of the mouse brain. These regions were chosen as they are believed to be homologous across species based on existing literature (see methods and *Supplementary file 1*). We next compared connectivity fingerprints from the three striatal seed regions in an independent mouse rsfMRI dataset, extracting rsfMRI timeseries from the three striatal seeds and correlating these with timeseries extracted from the twelve target regions (*Figure 2*). Analysis of the connectivity fingerprints (permutation testing of Manhattan distance – see Materials and methods) demonstrated that these three fingerprints were all significantly different from one another (CP.m vs NAcc: Distance = 2.57, p<0.001; CP.m vs CP.l: Distance = 6.67, p<0.001; NAcc vs CP.l: Distance = 6.15, p<0.001). This confirms that we can extract unique connectivity patterns in the mouse striatum and thus have isolated suitable seeds and targets for testing similarities and differences in connectivity fingerprints across

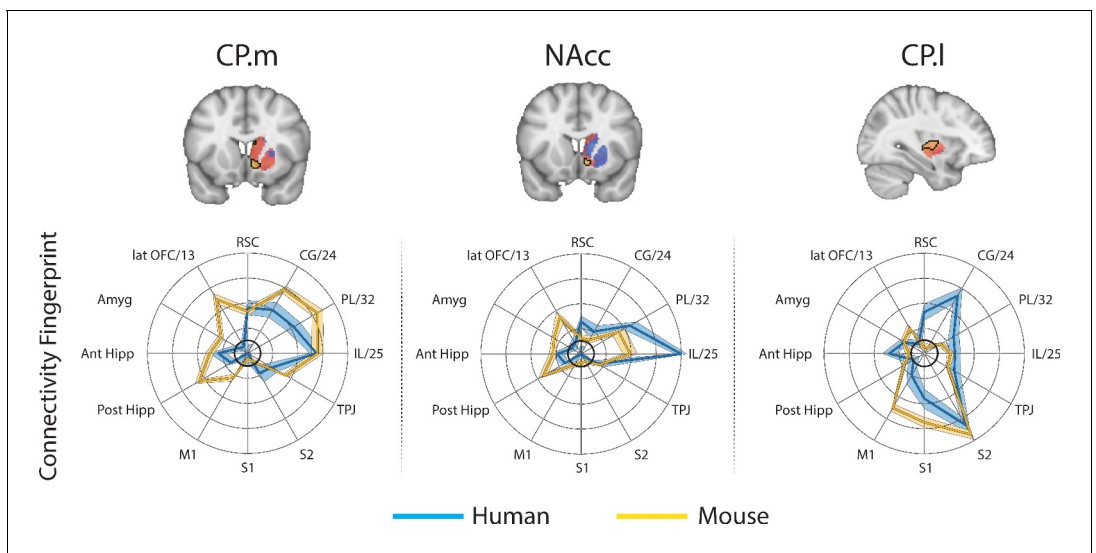

**Figure 2.** Brain images show unthresholded striatal t-maps of mouse-human similarity for CP.m, NAcc, and CP.l. Red-yellow voxels indicate increasingly positive correlations of connectivity fingerprints across species, blue-cyan voxels indicate increasing negative correlations of connectivity fingerprints across species. Black outlines indicate voxels that showed statistically significant similarity (TFCE p<0.05). Human and mouse connectivity fingerprints are shown underneath each brain image to highlight similarity in the connectivity pattern across species. Shaded error bars show the standard error of the mean. The data ranges from Z-values −0.1–0.4 and the thick grey circle shows 0.

The online version of this article includes the following figure supplement(s) for figure 2:

**Figure supplement 1.** Images show axial slices through the human striatum including the unthresholded t-maps highlighting the similarity in mouse-human connectivity fingerprints for CP.m.

**Figure supplement 2.** Images show axial slices through the human striatum including the unthresholded t-maps highlighting the similarity in mouse-human connectivity fingerprints for NAcc.

**Figure supplement 3.** Images show axial slices through the human striatum including the unthresholded t-maps highlighting the similarity in mouse-human connectivity fingerprints for CP.l.

**Figure supplement 4.** Brain images show unthresholded striatal t-maps of mouse-human similarity for CP.m, NAcc, and CP.l.

species. Voxel-wise correlation maps for these three seeds can also be seen in *Supplementary file 1*.

## Mouse to human comparison

These three mouse striatal seeds are commonly considered to reflect associative (CP.m), limbic (NAcc), and motor circuits (CP.l) (*Freeze et al., 2013*; *Gunaydin et al., 2014*; *Kiyokawa et al., 2012*). As such, we predicted that corresponding regions in the human striatum will exist within the human caudate nucleus, NAcc, and posterior putamen respectively. To test this, we extracted connectivity fingerprints for each voxel in the human striatum (defined as Harvard-Oxford subcortical atlas >33% threshold Caudate Nucleus, NAcc, and Putamen) and statistically compared each voxel's connectivity fingerprint with the three mouse connectivity fingerprints (*Figure 1*). This approach produced three voxelwise maps (Fishers r-to-z transformed maps) for each participant (one for each mouse fingerprint) illustrating voxelwise similarity with each mouse cortico-striatal circuit. These maps were input into a GLM (one sample t-test) for permutation testing and correction for multiple comparisons (TFCE $p<0.05$). All the unthresholded statistical maps in the section can be viewed at https://neurovault.org/collections/NFGTNVFX/. Voxels within the human NAcc possessed a statistically similar connectivity fingerprint with both the mouse NAcc and CP.m, whereas connectivity fingerprints within human posterior putamen voxels were statistically similar to the mouse CP.l (see *Figure 2*). *Figure 2* also highlights that even though there was an overlap in statistically similar voxels for mouse NAcc and CP.m in the human NAcc, there were clear subthreshold differences in the human caudate and putamen (positive correlations with CP.m but not NAcc) that distinguish these two striatal seeds. Using the task-based parcellation of *Pauli et al. (2016)*, we were able to assess the functional roles of the assigned human voxels. This showed that the human striatal voxels assigned to CP.m and NAcc were best localised to regions of the striatum processing stimulus value, whereas the human striatal voxels assigned to CP.l were localised to striatal regions contributing to motor control (see *Figure 2—figure supplement 4*).

Whilst this approach identified common striatal regions in humans and mice, there were a high number of voxels in the human striatum (85%) that did not show a significant correspondence to any of the three striatal connectivity fingerprints from the mouse – we refer to these as unassigned voxels. Unassigned voxels accounted for 85% of the caudate nucleus volume, 77% of the putamen volume, and 5% of the NAcc volume. Functionally, the unassigned voxels were localised to striatal regions associated with executive function, social/language, and action value (*Pauli et al., 2016*). These unassigned voxels could reflect the expansion of the prefrontal cortex in primates.

To further establish the cortical-striatal connectivity of the human regions identified above, we next used the resulting similarity t-maps to extract weighted timeseries and correlated them with the connectivity pattern of each voxel in the rest of the brain (*Figure 3*). Human CP.m voxels showed significant connectivity with frontal pole regions (Area 47 m), medial (RCZa, Area 23) and lateral prefrontal cortex (Area 46), anterior insula, supramarginal gyrus (hIP2), occipital pole (V1), cerebellar lobules HVI/Crus I and VIIIb, and hippocampus. The human homologue of NAcc also showed significant connectivity with anterior and middle cingulate cortex (Area 32 dl and area 32 p), occipital pole (V1), and cerebellar lobules Crus I and IX. The human homologue of CP.l showed significant connectivity with the middle frontal gyrus (Area 9/46), precentral gyrus (areas 6 and 4 p), SMA, Rolandic operculum (OP2), supramarginal gyrus (PFm), superior parietal lobule (area 7), precuneus (area 5), fusiform gyrus (FG2), and cerebellar lobule HVI. Full results tables are included in *Supplementary file 1*.

In order to highlight significant differences between humans and mice (rather than relying the absence of a significant effect), we performed a conjunction analysis between unassigned whole-brain connectivity maps and whole-brain connectivity maps from the three mouse seeds (see *Figure 3*). This analysis identified voxels in the human brain that showed significantly greater connectivity with unassigned voxels compared to all three human-mouse seeds independently (i.e. unassigned >CP .m and unassigned >NAcc and unassigned >CP .l). This analysis revealed striatal connectivity with the frontal pole (FPl and Area 46) that was unique to humans. It also revealed structures known for their connections with the prefrontal cortex: mediodorsal nucleus of the thalamus, and cerebellar lobules Crus I and Crus II. Finally, this analysis also refined the localisation of unassigned voxels within the human striatum. Whilst 85% of the human striatal voxels were not statistically similar to any of the three mouse cortico-striatal fingerprints, only 25.67% of those voxels were

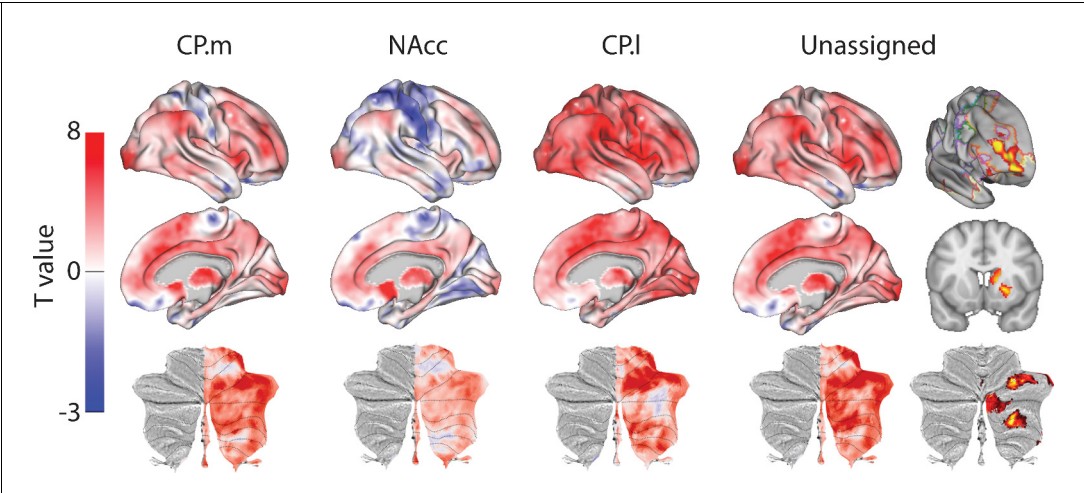

**Figure 3.** Unthresholded whole-brain connectivity maps showing regions interconnected with human homologs of CP.m, NAcc, CP.l, and unassigned voxels. The bottom row shows cerebellar activations on a flattened representation of the cerebellum and dotted black lines show the lobular boundaries (*Diedrichsen and Zotow, 2015*). The far-right column shows a thresholded conjunction analysis of voxels that possess significantly greater connectivity with unassigned voxels compared against all three mouse seeds. Outlines from the Yeo cortical parcellation highlight that the significantly different voxels in humans are principally in regions identified as the frontal -parietal network and the cerebellum.

significantly different. These were localised to two distinct striatal subregions; the first in the rostral caudate nucleus, the second in the anterior portion of the putamen. Both of these striatal regions are associated with executive functions and language processes. *Figure 3* shows that these unique cortico-striatal connections in humans also fall principally within the boundaries of the fronto-parietal network (FPN) as defined by *Yeo et al. (2011)*.

## Mouse to macaque comparison

We next applied the same procedure to compare cortico-striatal connectivity fingerprints in mice with cortico-striatal connectivity fingerprints in the macaque. Both species underwent similar rs-fMRI protocols including light sedation using anaesthesia, and as such this comparison allows us to account for the potential effects of anaesthesia on rs-fMRI connectivity. We hypothesised that a similar number of voxels will be unassigned in this comparison given that, from an evolutionary perspective rodents and primates diverged around 89 M years ago (*Cao et al., 2000*).

For each voxel in the macaque striatum (defined as caudate nucleus, NAcc, and putamen maps from the INIA19 macaque atlas; *Rohlfing et al., 2012*), we extracted a connectivity fingerprint using the same twelve targets. *Figure 4* shows that once again, we found regions of significant similarity, specifically the CP.m and NAcc showed significant similarity with voxels in the macaque NAcc, caudate head and caudate tail. The CP.l showed significant overlap with posterior regions of the putamen. This left 69% of voxels in the macaque striatum unassigned, that is, that did not show significant similarity with any mouse connectivity fingerprints. Unassigned voxels accounted for 79% of the caudate nucleus volume, 62% of the putamen volume, and 9% of the NAcc volume.

*Figure 5* displays regions showing significant connectivity with CP.m, NAcc, and CP.l voxels. CP.m voxels showed significant connectivity with the dorsomedial prefrontal cortex (Areas 9 m), premotor cortex (F2, F5), posterior lateral prefrontal cortex (Area 45b), anterior cingulate cortex (Area 24), posterior cingulate cortex (Area 23b), intraparietal sulcus (LIPd), cortex of the superior temporal sulcus and visual areas (V1 and V2). Significant connectivity with subcortical structures included the amygdala and caudate nucleus. Regions showing significant connectivity with NAcc voxels included anterior cingulate cortex (Area 24), premotor cortex (F4 and F5), posterior lateral prefrontal cortex (area 44), cortex of the superior temporal sulcus, and visual cortex (V2). Subcortical connectivity included the macaque NAcc, amygdala, and hippocampus. Regions showing significant connectivity with CP.l voxels included the premotor cortex (F2, F5), anterior cingulate cortex (area 24) and posterior cingulate cortex (Areas 23 and 31), somatosensory cortex on the posterior bank of central sulcus,

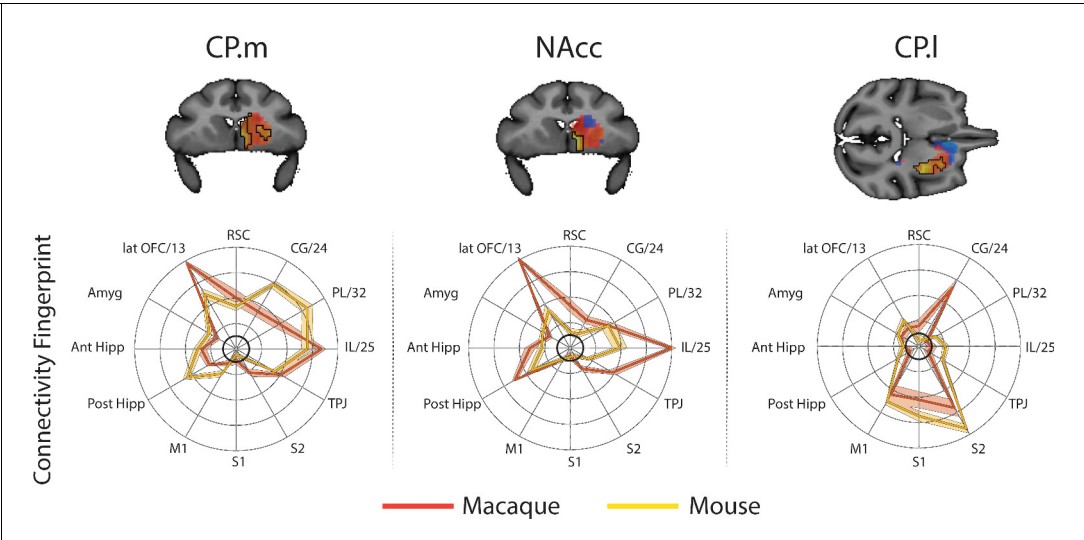

**Figure 4.** Brain images show unthresholded striatal t-maps of mouse-macaque similarity for CP.m, NAcc, and CP.l. Red-yellow voxels indicate increasingly positive correlations of connectivity fingerprints across species, blue-cyan voxels indicate increasing negative correlations of connectivity fingerprints across species. Black outlines indicate voxels that showed statistically significant similarity (TFCE p<0.05). Macaque and mouse connectivity fingerprints are shown underneath each brain image to highlight the similarity in the connectivity pattern across species. Shaded error bars show the standard error of the mean. The data ranges from Z-values −0.1 to 0.4 and the thick grey circle shows 0.

The online version of this article includes the following figure supplement(s) for figure 4:

**Figure supplement 1.** Images show axial slices through the macaque striatum including the unthresholded t-maps highlighting the similarity in mouse-macaque connectivity fingerprints for CP.m.

**Figure supplement 2.** Images show axial slices through the macaque striatum including the unthresholded t-maps highlighting the similarity in mouse-macaque connectivity fingerprints for NAcc.

**Figure supplement 3.** Images show axial slices through the macaque striatum including the unthresholded t-maps highlighting the similarity in mouse-macaque connectivity fingerprints for CP.l.

and Intraparietal sulcus (LIP). Subcortical regions included the putamen (largest activation) and the Caudate nucleus.

The conjunction analysis (i.e. unassigned >CP .m and unassigned >NAcc and unassigned >CP .l) highlighted regions of significantly different in macaques compared to mice. These included dorso-lateral prefrontal cortex (Area 46d), premotor cortex (F2), anterior insula, cortex of the superior

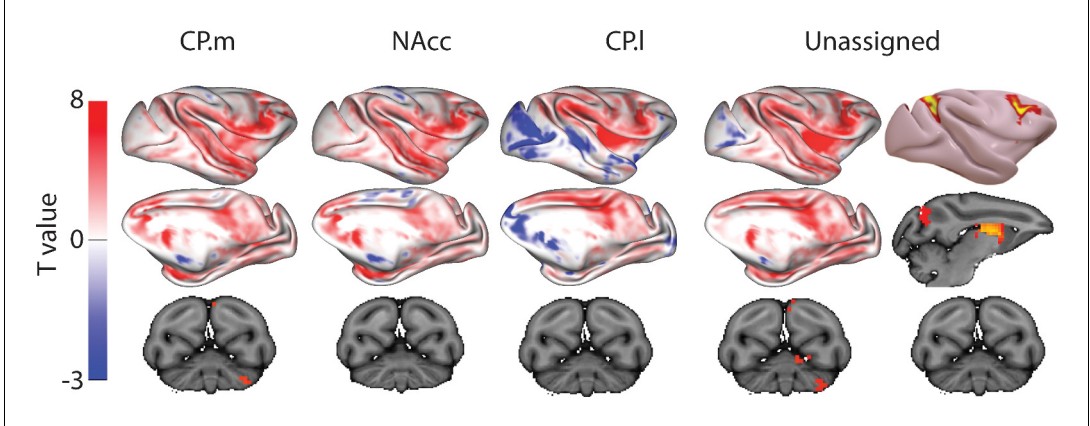

**Figure 5.** Unthresholded whole-brain connectivity maps showing regions interconnected with CP.m, NAcc, CP.l, and unassigned voxels. The bottom row shows cerebellar activations. The far-right column shows a thresholded conjunction analysis of voxels that possess significantly greater connectivity with unassigned voxels compared against all three mouse seeds.

temporal sulcus, inferior parietal lobule (Area 7A), and visual cortex (V1). Subcortical differences were seen in both the rostral caudate nucleus and putamen as in humans. The application of the conjunction analysis showed that only 20% of striatal unassigned voxels were significantly different from all three mouse seeds. These significantly different voxels accounted for 34% of the caudate nucleus, 10% of the putamen, and 1% of NAcc volume (four voxels), suggesting that differences between macaques and mice were largely driven by differences in the caudate nucleus - as in the previous mouse-human analyses. Full results tables are included in *Supplementary file 1*.

## Macaque to human comparisons

In order to determine whether some striatal connectivity features of the unassigned voxels could be reflecting uniquely human specialisations, we applied the same protocols to compare striatal connectivity fingerprints in humans and macaques. Similar to what was done for the mouse, macaque caudate body, NAcc, and putamen were defined as seeds using connectivity-based parcellation (see Materials and methods). We used two target models; 1) the original targets used in the mouse model and 2) an extended model that included additional regions in the lateral and medial PFC that have been shown previously to have homologous connectivity fingerprints across macaques and humans (*Neubert et al., 2014*; *Sallet et al., 2013*). This included area 9/46d, area 9/46 v, Area 44, area FPm, and SMA. Connectivity fingerprints for all three seeds were independent in both the mouse model (Caudate vs NAcc: Distance = 3.51, p<0.001; Caudate vs Putamen: Distance = 3.62, p<0.001; NAcc vs Putamen: Distance = 4.96, p<0.001) and the extended model (Caudate vs NAcc: Distance = 4.77, p<0.001; Caudate vs Putamen: Distance = 5.32, p<0.001; NAcc vs Putamen: Distance = 6.12, p<0.001).

For each voxel of the human Caudate Nucleus, NAcc, and Putamen (based on Harvard Oxford subcortical atlas >33% threshold), we extracted the human connectivity fingerprint and compared it to connectivity fingerprints for the macaque caudate body, NAcc, and putamen. Unsurprisingly, the human-macaque comparisons were more closely aligned then the mouse-human comparisons. Specifically, the human voxels in the caudate nucleus were significantly similar to the macaque caudate body, human voxels in the NAcc were significantly similar to the macaque NAcc, and human voxels in the putamen were significantly similar to the macaque putamen (see *Figure 6*). Using the task-based parcellation of *Pauli et al. (2016)*, the human voxels assigned to the macaque caudate body overlapped with striatal regions dedicated to executive function, social/language processes, and action value. The human voxels assigned to the macaque NAcc overlapped with striatal regions dedicated to stimulus value processing. The human voxels assigned to the macaque putamen best overlapped with striatal motor control regions (see *Figure 6—figure supplement 4*).

The human-macaque comparison produced fewer unassigned voxels than the human-mouse comparison, with only 31% of striatal voxels unassigned using the same model used in mice, and 20% unassigned using the extended model. These voxels were localised anatomically to the NAcc (30% of NAcc voxels unassigned using the reduced mouse model, 24% using the extended model) and Caudate (35% of caudate voxels unassigned using the reduced mouse model, 23% using the extended model), and showed functional association with striatal regions associated with executive functions. Although 30/23% of unassigned voxels were localised in the NAcc, the conjunction analysis isolating significant differences between species showed that only voxels in the caudate nucleus were significantly different between species.

As previously, we extracted weighted timeseries from each similarity t-map to investigate voxelwise connectivity (*Figure 7*). The human-macaque caudate nucleus maps showed significant connectivity with mid cingulate cortex (RCZa, CCZ) and middle (Area 46) and inferior frontal gyri (pars triangularis), inferior parietal lobule (hIP3), precuneus (Area 7 p), occipital pole (V1 and V3), cerebellar lobules HVI and Crus II. Human-macaque NAcc maps showed significant connectivity with the anterior cingulate cortex (area 32pl) and cerebellar lobule Crus IX. Human-macaque putamen maps showed significant connectivity with regions in the mid cingulate cortex (RCZa) and lateral prefrontal cortex (area 46) and premotor cortex (area 6), supramarginal gyrus (area PF), middle temporal gyrus, occipital lobe (V3), and cerebellar lobule HVI.

When comparing connectivity differences between assigned vs unassigned voxels using the conjunction analysis, the unassigned voxels showed significant connectivity with frontal pole (FPl), precuneus, occipital pole (V1), and subcortical structures known for projecting to the prefrontal cortex (the mediodorsal nucleus of the thalamus and the cerebellar lobule Crus I). Coupling with the FPl

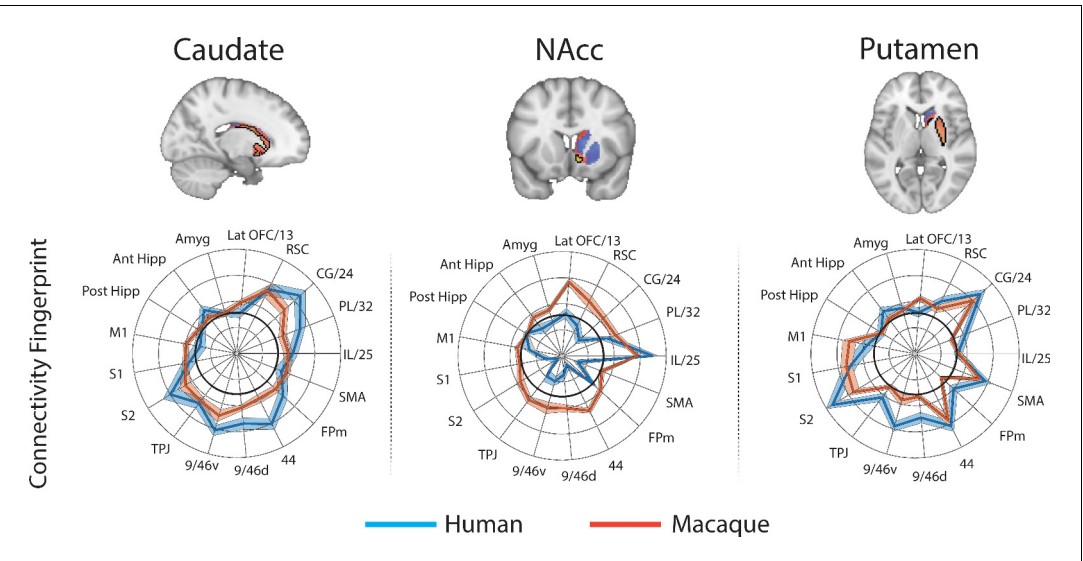

**Figure 6.** Brain images show unthresholded striatal t-maps of macaque-human similarity for caudate body, NAcc, and putamen. Red-yellow voxels indicate increasingly positive correlations of connectivity fingerprints across species, blue-cyan voxels indicate increasing negative correlations of connectivity fingerprints across species. Black outlines indicate voxels that showed statistically significant similarity (TFCE p<0.05). Human and macaque connectivity fingerprints are shown underneath each brain image to highlight the similarity in connectivity pattern across species. Shaded error bars show the standard error of the mean. The data ranges from Z-values −0.25–0.4 and the thick grey circle shows 0.

The online version of this article includes the following figure supplement(s) for figure 6:

**Figure supplement 1.** Images show axial slices through the human striatum including the unthresholded t-maps highlighting the similarity in macaque-human connectivity fingerprints for caudate body.

**Figure supplement 2.** Images show axial slices through the human striatum including the unthresholded t-maps highlighting the similarity in macaque-human connectivity fingerprints for NAcc.

**Figure supplement 3.** Images show axial slices through the human striatum including the unthresholded t-maps highlighting the similarity in macaque-human connectivity fingerprints for putamen.

**Figure supplement 4.** Brain images with unthresholded striatal t-maps showing macaque-human similarity for Caudate Nucleus, NAcc, and Putamen.

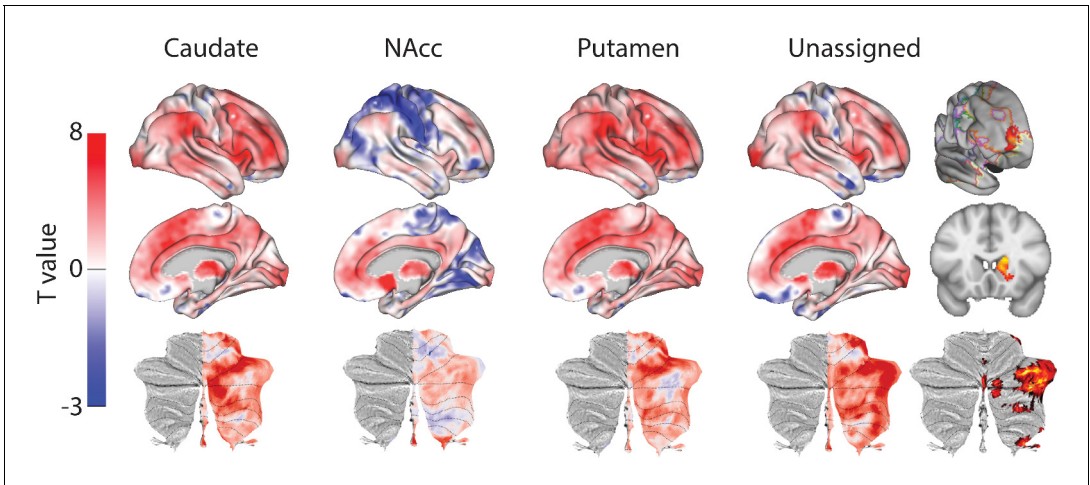

**Figure 7.** Unthresholded whole-brain connectivity maps showing regions interconnected with caudate body, NAcc, putamen, and unassigned voxels. The bottom row shows cerebellar activations on a flattened representation of the cerebellum and dotted black lines show the lobular boundaries (**Diedrichsen and Zotow, 2015**). The far-right column shows a thresholded conjunction analysis of voxels that possess significantly greater connectivity with unassigned voxels compared against all three mouse seeds. Outlines from the Yeo cortical parcellation highlight that the significantly different voxels in humans are principally in regions identified as the frontal-parietal network and the cerebellum.

could reflect the expansion of the lateral frontal pole since the last common ancestor to humans and macaques (*Neubert et al., 2014*). The number of unassigned voxels was similar to the number of voxels that were significantly different (31% unassigned voxels but 18.1% of them were significantly different to our three fingerprints). Although unassigned voxels were localised to both the NAcc and caudate nucleus, only voxels in the caudate nucleus were significantly different in humans compared to macaques. These have been localised to regions of caudate connected with the FPN that process executive functions and action value (*Choi et al., 2012*; *Pauli et al., 2016*). As with mice, this unique cluster of voxels was co-localised with FPN as defined by *Yeo et al. (2011)*. All the statistical maps reported in the section can be viewed at https://neurovault.org/collections/NFGTNVFX/.

## Discussion

As the use of mouse models for studying function and disease rapidly increases in neuroscience, it is crucial to develop methods that can harmonise results across species. Here, for the first time, we have used rsfMRI as a common methodology to establish similarities and differences in striatal-cortical organisation in humans, macaques, and mice. Using our connectivity fingerprint matching approach, we could identify NAcc consistently across species making it a reliable target for translational neuroscience. Although portions of the caudate nucleus and putamen in both humans and macaques showed similar connectivity fingerprints with the mouse, there were also large regions of the human and macaque striatum that appeared to be unique and unassigned. Regions of significant difference across species appear to be mostly localised to the anterior putamen and caudate body. In both human-mouse and human-macaque comparisons, unassigned voxels showed significantly greater connectivity with the lateral frontal pole (Area 46 in mice and FPl in both mice and macaques) and prefrontal-projecting subcortical structures including the dorsomedial nucleus of the thalamus and cerebellar lobules Crus I and Crus II.

### Striatum

Consistent with previous parcellations of the mouse striatum, our connectivity based parcellation using Allen tracer data identified three regions with unique connectivity fingerprints; the NAcc, medial and lateral caudoputamen (CP.m and CP.l). Previous studies have suggested that these regions map on to the distinct functional domains of limbic, association, and sensorimotor, respectively (*Balleine et al., 2009*; *Hunnicutt et al., 2016*; *Thorn et al., 2010*; *Yin and Knowlton, 2006*). Our results suggest that the connectivity fingerprint of the NAcc is highly conserved across mice and primates. Human and macaque striatal voxels assigned to the mouse NAcc, and the human voxels assigned to the macaque NAcc, were all discretely localised within the boundaries of the human NAcc as defined by the Harvard-Oxford subcortical atlas and INIA atlas in primates (see *Figure 8*). This is consistent with tracer studies by *Mailly et al. (2013)* and *Heilbronner et al. (2016)* who also demonstrated an overlap in NAcc connectivity fingerprints across rats and non-human primates. This conserved brain network could underpin their similar functional role in motivation and reinforcement

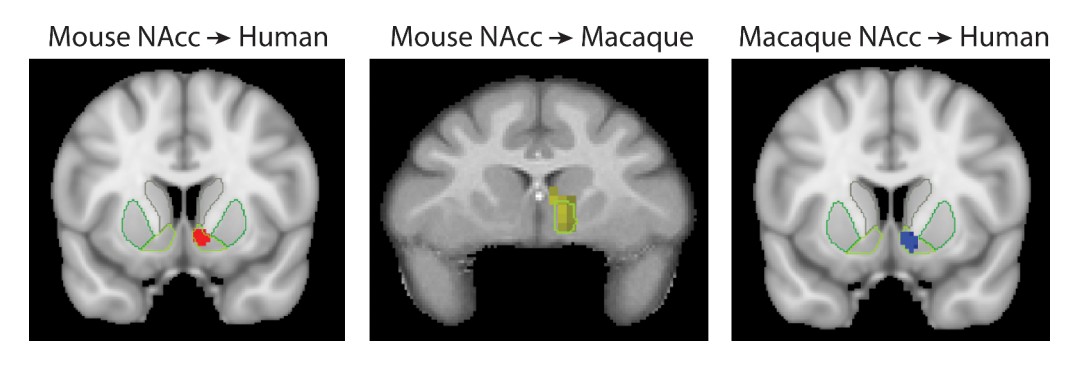

**Figure 8.** Brain images showing the spatial overlap of the NAcc across species. Voxels highlight regions of statistically significant similarity across species (TFCE p<0.05). The light green outline in each image shows the anatomical boundaries of the human and macaque NAcc.

learning. For example, lesions to the rat NAcc have been shown to alter performance in value-based decision-making paradigms such as delayed discounting tasks (*Cardinal et al., 2001*). Similarly, non-invasive imaging studies in humans have linked individual variability in NAcc activity to individual variability in delayed discounting preferences, mirroring what has been shown in rats (*Hariri et al., 2006*). Aberrant performance on value-based decision-making paradigms, as well as altered NAcc activity, has been proposed as a hallmark for a number of psychiatric conditions including Obsessive Compulsive Disorder, addiction, schizophrenia and others (*Gunaydin and Kreitzer, 2016*). Given the high degree of conservation across species in NAcc connectivity, we would suggest that the NAcc is a reliable translational target for mouse models of psychiatric conditions.

The CP.m and CP.l are believed to contribute to associative and sensorimotor motor processes respectively and would thus be expected to correspond to regions of the caudate and putamen with similar functional properties (*Balleine et al., 2009*; *Yin and Knowlton, 2006*). Our results showed that voxels in the posterior segment of the putamen in both humans and macaques shared a significantly similar connectivity fingerprint with the mouse CP.l (see *Figures 2* and *4*), suggesting that sensorimotor cortico-striatal circuits could also be comparable across species. This is crucial for translational models of sensorimotor deficits such as Parkinson's disease. However, 85% of voxels in the human striatum failed to show significant similarity with any of the three mouse striatal seeds. These unassigned voxels were localised to the caudate nucleus and putamen. The conjunction analysis highlighted voxels that were significantly different in humans compared to mice (as opposed to voxels that failed to reach the statistical significance threshold). This approach confirmed that 25% of voxels in human striatum – specifically the anterior portion of the putamen and caudate body - possess significantly different connectivity fingerprints in humans compared to mice. These regions of the striatum have been shown to receive projections from the lateral prefrontal cortex in both humans and non-human primates (*Alexander et al., 1986*; *Choi et al., 2017*; *Verstynen et al., 2012*), and our comparison with task-based parcellations of the striatum suggest that these regions process action value, executive functions, and social and language processes in a meta-analysis of over 10,000 human fMRI studies (*Pauli et al., 2016*). Although these results could reflect the effects of anaesthesia on rs-fMRI in mice, we believe this is unlikely given that 1) significant similarities were found between humans and mice for limbic (NAcc) and sensorimotor (CP.l) cortico-striatal networks, and 2) rs-fMRI data collected from macaques used a similar anaesthesia protocol to that of mice and yet the macaque caudate nucleus showed significant similarity with most voxels in the human caudate nucleus (*Figure 6*). Rather, these results are likely to reflect evolutionary pressures acting upon brain circuits underpinning cognitive behaviours (*Passingham and Wise, 2012*). It has been argued that evolution has led to increased cognitive development particularly in the primate and human lineages, whereas human motor and appetitive behaviours are more conserved across species (*Murray et al., 2016*). As such, one would predict that limbic (NAcc) and sensorimotor (CP.l) cortico-striatal circuits would be largely conserved, whereas associative cortico-striatal circuits would have changed most across species. We suggest that these differences between human and mouse striatal fingerprints are more likely to reflect differences in connectivity with regions of the lateral prefrontal cortex that likely underpin human behaviours that have developed most with evolution, that is executive functions, and social/language abilities. As such we would caution researchers using mouse models for disorders predominantly affecting these processes as there appears to be less clear translation between mice and humans.

## Prefrontal cortex

Most theories on the functions of the prefrontal cortex have been derived from studies in humans and nonhuman primates, awaiting translation in rodents. Multiple approaches have been taken to establish homologs across species, including similarities in cytoarchitecture, similarities in connectivity, and equivalence of function. While each of these approaches has been used to debate the existence of the granular PFC in rodents, there is at least increasing consensus that mouse prefrontal regions IL, PL, and Cg could be equivalent to human and macaque areas 25, 32, and 24 respectively (*Bicks et al., 2015*). *Heilbronner et al. (2016)* showed similar striatal projection patterns in rats and macaques using tracers from IL/25, PL/32, and Cg/24 in both these species. However, there still is quite some debate on whether other parts of human prefrontal cortex are present in the rodent (*Laubach et al., 2018*). At the very least, human prefrontal cortex as extended substantially in absolute terms, although its relative extension compared to other primates is also a matter of fierce

debate (*Barton and Venditti, 2013*; *Passingham and Smaers, 2014*). For the definition of our connectivity fingerprints, we have only used regions whose homology across species has been well established in the literature (see Materials and methods and *Supplementary file 1* tables 1-3). However, our results speak to the debate about translational results in prefrontal cortex in two ways.

First, and most obvious, we find voxels in the human striatum that possess a connectivity fingerprint that is not found in either the mouse or macaque. These voxels all tended to have a strong functional connectivity with, among others, parts of the human lateral prefrontal cortex (area 46 and lateral frontal pole (FPl)). Although area 46 has a clear homolog in the macaque, it is part of granular dorsolateral prefrontal cortex which is believed to be an anthropoid primate specialisation (*Passingham and Wise, 2012*; *Preuss and Goldman-Rakic, 1991*) and therefore not present in the mouse. The lateral part of the frontal pole has been identified in humans based on both cytoarchitecture and connectivity (*Bludau et al., 2014*), but a macaque homolog is not detectable using rs-fMRI raising the possibility that this is a human or at least ape specialisation (*Neubert et al., 2014*). The fact that the unassigned striatal voxels predominantly show connectivity with these areas suggests the presence of a unique cortical-striatal loop. Based on our findings, this loop would be mostly involved in higher order human behaviours, as are associated with FPl (*Hartogsveld et al., 2018*; *Vendetti and Bunge, 2014*), suggesting that these might be difficult to study using the translational paradigm.

Our findings also warrant a second caution with regard to the translation of prefrontal results. Even though human striatal regions possessed similar connectivity fingerprints to those of mice, it is possible that similar circuits could include novel projections in humans (*Mars et al., 2018*). For instance, voxels in the human striatum assigned to CP.m showed the expected connectivity with homologous target regions such as the medial frontal cortex, but also with regions in the dorsal prefrontal cortex (dlPFC) and inferior parietal lobule (*Figure 3*). Since it's disputed whether homologs of dlPFC and inferior parietal lobule exist within mice, it seems likely the CP.m network is embedded within a larger network in humans that includes novel regions. We therefore urge caution interpreting translational results: even if regions of interest are homologous, they might be embedded in larger networks that include novel areas.

## Cerebellum

Humans showed unique striatal connectivity with parts of association cortex, including the anterior prefrontal cortex, but also with large regions of cerebellar cortex. Specifically, the cerebellar lobules showing unique striatal connectivity in humans were Crus I and Crus II, which have been shown to be interconnected with the prefrontal cortex (*Kelly and Strick, 2003*; *O'Reilly et al., 2010*), and contribute to cognitive processes including rule-guided behaviour (*Balsters et al., 2013*; *Balsters and Ramnani, 2011*; *Balsters and Ramnani, 2008*) and language (*Lesage et al., 2017*; *Lesage et al., 2012*; *Mariën et al., 2014*). Tracer studies in rats and non-human primates have shown that prefrontal projections to the cerebellar cortex are conserved across species (*Kelly and Strick, 2003*; *Schmahmann and Pandya, 1997*; *Wiesendanger and Wiesendanger, 1982*); however, research also suggests that prefrontal-cerebellar circuits have selectively expanded in humans compared to other species. Specifically, prefrontal projections to the pontine nucleus (*Ramnani et al., 2006*), volume of prefrontal-projecting cerebellar lobules Crus I and Crus II (*Balsters et al., 2010*; *Luo et al., 2017*), and the volume and connections of the dorsal dentate nucleus (*Baizer, 2014*; *Matano, 2001*; *Steele et al., 2016*) have all expanded in humans relative to motor cerebellar circuitry across species. Studies of cerebellar evolution have generally suggested that Crus I and Crus II are homologous across species (*Larsell, 1952*) even though they have expanded in humans. The findings of this study could suggest that along with selective expansion, there may be additionally novel cortico-cerebellar connections in humans that have contributed to the expansion of the cortico-cerebellar system in humans. Given that the Area FPl homolog in non-human primates and mice is unclear or absent, it is plausible that this region has generated novel projections to both the striatum and cerebellum. The finer grained 17-Network parcellation of the human cerebral cortex by *Yeo et al. (2011)*, includes a region similar to Area FPl identified in this study (Network 13). *Buckner et al. (2011)* analysis of cerebellar connectivity using the Yeo cortical parcellation shows that this region projects to Crus I and Crus II, adding further evidence to suggest that the some of the expansion of Crus I and Crus II could be due to novel inputs from the frontal pole in humans.

## Caveats and cautions

Here, we used rsfMRI as a common methodology to compare connectivity fingerprints across humans, nonhuman primates, and mice. However, it is important to acknowledge the limitations of rsfMRI for making inferences about connectivity compared to tracer-based methods or diffusion weighted imaging (DWI)-based tractography. Tracer-based methods have often been considered to be the 'gold standard' for connectivity research, and comparative neuroanatomy. The level of detail available using tracer methods goes far beyond what is currently possible using fMRI. For example, tracer-based methods illustrate direction-specific monosynaptic connections, something that is not currently available using rsfMRI. When interpreting the results presented here, it is important to consider that differences across species could reflect network level differences as opposed to specific changes in monosynaptic connections (*Mars et al., 2018*; *Thiebaut de Schotten et al., 2019*). For example, in both mouse-human and macaque-human analyses, we found increased functional connectivity with unassigned voxels in the anterior putamen/caudate and the occipital lobe. It is unclear whether this reflects a polysynaptic network effect or whether this reflects a change in monosynaptic connections between the occipital lobe and basal ganglia previously shown in mice (*Hunnicutt et al., 2016*; *Khibnik et al., 2014*) and primates (*Saint-Cyr et al., 1990*; *Schmahmann and Pandya, 2006*). Tracer-based methods can also provide details regarding cell-specific connectivity that distinguishes between different classes of interneurons. *Ährlund-Richter et al. (2019)* recently produced a whole brain atlas of cell-specific connections to medial PFC in mice, providing greater insight into the unique structure-function relationships that support information processing within the medial PFC. Although such methods provide exquisite levels of neuroanatomical detail, the lack of comparable methods in humans makes it difficult to translate these findings across species. While rsfMRI is unlikely to ever reach the level of specificity achieved by tracer methods, we believe the ability to use this technique in all species makes it a powerful tool for comparative neuroanatomy.

DWI is another technique that can be used across species as an alternative to rsfMRI. Previous studies have used DWI with great effect to compare human and nonhuman primate brain connectivity (*Folloni et al., 2019*; *Mars et al., 2018*); however, DWI has proven less successful in rodents. *Calabrese et al. (2015)* and *Sinke et al. (2018)* both found low similarity between DWI and tracer-based tracts in mice and rats, respectively. In comparison, *Grandjean et al. (2017)* found 87–88% overlap between rsfMRI connectivity and tracer connectivity in cortico-cortico circuits and cortico-striatal circuits. The strong agreement between the modalities is perhaps surprising, since rsfMRI connectivity and studies on neuronal tracers are fundamentally different. Anterograde viral tracers, such as those used for the Allen Mouse Brain Connectome, are unidirectional, follow individual axons and do not cross synapses. In contrast, rsfMRI is bidirectional and is not necessarily constrained by synapses. For example, the connections present in rsfMRI but missing from the anterograde tracer data are believed to be false positives, but they could also be real retrograde connections or real polysynaptic connections. This once again highlights one of the difficulties in comparing rsfMRI data with tracer studies. It's often unclear whether differences across species are due to evolutionary changes in neuroanatomy or differences in methodologies. As such, it's crucially important to compare like-with-like using the same methods across species. As nonhuman data sharing repositories grow, it will also be possible to address further interesting questions regarding gender differences (currently all the mice and macaque data were male).

One caveat to the approach used here is the definition of striatal seeds in the mouse. Data from both tracers and rsfMRI suggested that a three cluster solution best explained cortico-striatal connectivity in mice. While many studies in mice refer to a three domain striatal model (i.e. associative, limbic, motor), this concept is largely based upon research in non-human primates and rats (*Balleine et al., 2009*; *Thorn et al., 2010*; *Yin and Knowlton, 2006*). Although some anatomical studies in mice are providing greater support for the three domain model (*Hunnicutt et al., 2016*), there is a need for more operant studies in mice and more large scale anatomical studies in rats. It is also possible that whilst the three cluster striatal solution is the most consistent across subjects (mice), this may be an oversimplification of the cortico-striatal system. Studies in humans using rsfMRI connectivity have found that the caudate nucleus can be segmented into 1–9 regions, whilst the putamen can be segmented into 1–6 clusters (*Choi et al., 2012*; *Janssen et al., 2015*; *Jaspers et al., 2017*; *Jung et al., 2014*; *Tian et al., 2020*). The recent work of *Tian et al. (2020)*,

used a novel gradient-based approach with 1000 rsfMRI datasets at 3T and 7T. Their results highlight the various scales of resolution, ranging from scale I (1 putamen, 1 caudate, and 1 NAcc) to scale III (4 putamen, 4 caudate, 2 NAcc subdivisons). However, MRI based parcellations of the striatum appear to be much coarser than those seen in mice using tracer methods. For example, *Hintiryan et al. (2016)* utilised a novel neuroanatomical approach to map the cortico-striatal projectome, identifying 29 striatal regions with distinct connectivity fingerprints, and *Chon et al. (2019)* further extended this to present a 33 striatal cluster solution. Even using state of the art fMRI acquisition and analysis methods, the limitations of fMRI and its spatial resolution make a detailed analysis such as that of *Hintiryan et al. (2016)* and *Chon et al. (2019)* unfeasible. Advances in MRI methods may make it possible to investigate higher resolution parcellations of the striatum and other cortical and subcortical structures in the future, which may yield greater correspondence between mice and human brain circuits. We note, however, that the current data are of the best resolution and quality currently in general use (*Grandjean et al., 2020*; *Glasser et al., 2016*; *Milham et al., 2018*) and it is likely most translational studies will not be working with data of superior quality.

As well as seed definition, connectivity fingerprint matching relies on the accurate definitions of target areas across species. Indeed, connectivity fingerprint matching is heavily reliant on the fact that 1) connectivity strength will differ across targets, and 2) targets are anatomical and functional homologs across species. Here, we chose to focus on cortico-striatal circuits because their connectivity has been well-defined in humans, nonhuman primates, and mice (*Alexander et al., 1986*; *Haber, 2016*; *Hintiryan et al., 2016*; *Verstynen et al., 2012*). In addition, we selected targets that have already been used in previous studies comparing humans with macaques (*Mars et al., 2013*; *Mars et al., 2011*; *Neubert et al., 2015*; *Neubert et al., 2014*; *Sallet et al., 2013*), and have been shown project to different striatal regions. This produced our extended primate target model (*Supplementary file 1* tables 2,3) of 17 targets which are broadly categorised as cingulate, orbitofrontal, subcortical, motor, temporal, ventrolateral PFC, dorsolateral PFC, SMA, and frontal pole. Given the debate surrounding mouse homologs of the lateral PFC, frontal pole, and SMA, these targets were not included in our reduced mouse model (see *Supplementary file 1* table 1). However, it is important to note that in our human-macaque analysis, there was very little difference in the results generated using the reduced vs extended target models, suggesting that the results based upon the reduced mouse model are stable and not impacted by the addition of extra targets. As this line of research continues, it will be possible to refine our target models based on the results of previous studies.

## Conclusions

Here, we demonstrate the potential of connectivity fingerprint matching to bridge the gap between mouse and primate neuroanatomy. Our results highlight the core properties of a mouse to primate striatum, including similarities in connectivity fingerprints for NAcc across species that could be a useful model for translational neuroscience. However, we would caution against researchers comparing medial and lateral regions of the mouse caudoputamen with the primate caudate nucleus and putamen. Although homologs were identified, there were also clear differences in connectivity patterns that require further investigation. We propose that these differences reflect the expansion of frontal cortex in primates, along with the relative expansion of area FPl in humans. These results will hopefully add to the on-going debates surrounding similarities in cortical brain regions across species, that is the existence of the PFC in mice. Further studies using connectivity fingerprint matching could help to refine where similarities and differences exist across species in other brain structures including medial prefrontal cortex and orbitofrontal regions.

## Materials and methods

### Targets and seeds
#### Reduced/mouse model
The reduced model includes twelve target regions common to all species: 1) Infralimbic (Area 25), 2) Prelimbic (Area 32), 3) Cingulate areas (Area 24), 4) Retrosplenial Cortex (Area 30), 5) Lateral Orbitofrontal Cortex (Area 13), 6) Basolateral Amgydala, 7) Dorsal (Anterior) Hippocampus, 8) Ventral (Posterior) Hippocampus, 9) Primary Motor Cortex (M1), 10) Primary Somatosensory Cortex (S1), 11)

Supplemental Somatosensory Cortex (S2), 12) Temporal association area (TPJp). Targets were 3 × 3×3 voxels in all species. Further details can be found in *Supplementary file 1* tables 1-3.

### Extended/primate model

The extended model includes the twelve reduced model targets and five additional targets that are common to macaques and humans, but not mice: 13) Area 9/46d, 14) Area 9/46 v, 15) Area 44d, 16) FPm, 17) SMA. Targets were 3 × 3×3 voxels in all species. Further details can be found in *Supplementary file 1* tables 2 and 3.

### Mouse striatal seeds

Allen Institute database of tracer experiments were obtained and processed as described in *Grandjean et al. (2017)*. Briefly, viral-tracer maps were downloaded using the query form from the Allen Institute database. We limited the scope of our analysis to the injections carried in the Isocortex of wild-type C57BL/6 animals, with injection volume ranging from 0.12 to 0.4 µl. This resulted in 68 tracer experiments, which were resampled at 200 $\mu m^3$ and coregistered into the Allen Reference Atlas (v3) using Advanced Normalisation Tools (ANTS) (version 2.1, http://picsl.upenn.edu/software/ants/). The connectivity between striatal voxels and cortical seeds was determined after normalising the fluorescence with the volume of injection (for further details, see *Oh et al., 2014*).

We established connectivity strength (Z-transformed terminal tracer volume) between 68 injection sites in isocortex and each voxel in the caudoputamen, NAcc, and fundus. We then used connectivity-based parcellation to partition the mouse basal ganglia into regions with unique connectivity fingerprints (*Balsters et al., 2016*; *Balsters et al., 2018*). The optimal solution based on silhouette value was three regions with unique connectivity fingerprints based on anterograde tracers. These are labelled medial caudoputamen (CP.m), Nucleus accumbens (NAcc), and lateral caudoputamen (CP.l). The percent variance explained by the first eigenvariate for each seed was 57.72% ± 6.42, 49.18% ± 6.12, and 60.14% ± 6.89 for CP.m, NAcc, and CP.l, respectively.

### Macaque seeds

Connectivity-based parcellation was applied to the macaque resting state data in order to create seeds with unique connectivity fingerprints. The optimal solution based on silhouette value was a 5 cluster solution keeping the NAcc and putamen whole, and segmenting the caudate nucleus into three segments (the body and two segments in the tail of the caudate). We focussed our analysis on the caudate body, NAcc, and putamen given the small number of voxels contributing to the caudate tail seeds. The percent variance explained by the first eigenvariate for each seed was equivalent to that of mice - 56.14% ± 6.39, 51.83% ± 5.87, and 56.28% ± 7.56 for Caudate body, NAcc, and putamen, respectively.

## MRI data acquisition

### Mouse

Mouse fMRI and anatomical scans were collected from 20 wildtype C57BL/6J animals (males, median age = 82 days; median weight 26 grams). Animals were caged in standard housing (maximum 5 animals/cage), with food and water ad libitum, and a 12 h day/night cycle. All MRI scans were conducted in the light phase. Protocols for animal care, magnetic resonance imaging, and anaesthesia were carried out under the authority of personal and project licenses in accordance with the Swiss federal guidelines for the use of animals in research, and under licensing from the Zürich Cantonal veterinary office.

Anaesthesia was induced with 4% isoflurane and the animals were endotracheally intubated and the tail vein cannulated. Mice were positioned on a MRI-compatible cradle, and artificially ventilated at 80 breaths per minute, 1:4 $O_2$ to air ratio, and 1.8 ml/h flow (CWE, Ardmore). A bolus injection of medetomidine 0.05 mg/kg and pancuronium bromide 0.2 mg/kg was administered, and isoflurane was reduced to 1%. After 5 min, an infusion of medetomidine 0.1 mg/kg/hr and pancuronium bromide 0.4 mg/kg/hr was administered, and isoflurane was further reduced to 0.5%. The animal temperature was monitored using a rectal thermometer probe, and maintained at 36.5°C ± 0.5 during the measurements. The preparation of the animals did not exceed 20 min.

Data acquisition was performed on a Biospec 70/16 small animal MR system (Bruker BioSpin MRI, Ettlingen, Germany) with a cryogenic quadrature surface coil (Bruker BioSpin AG, Fällanden, Switzerland). After standard adjustments, shim gradients were optimized using mapshim protocol, with an ellipsoid reference volume covering the whole brain. For functional connectivity acquisition, a standard gradient-echo EPI sequence (GE-EPI, repetition time TR = 1000 ms, echo time TE = 15 ms, in-plane resolution RES = 0.22 × 0.2 mm$^2$, number of slice NS = 20, slice thickness ST = 0.4 mm, slice gap SG = 0.1 mm) was applied to acquire 2000 vol in 38 min. In addition, we acquired anatomical T2*-weighted images (FLASH sequence, in-plane resolution of 0.05 × 0.02 mm, TE = 3.51, TR = 522 ms). The levels of anaesthesia and mouse physiological parameters were monitored following an established protocol to obtain a reliable measurement of functional connectivity (*Grandjean et al., 2014*; *Zerbi et al., 2015*).

## Macaque

Macaque fMRI and anatomical scans were collected from 10 healthy macaque monkeys (*Macaca mulatta*, 10 males, median age = 4.98 years; median weight 9.25 kg). Protocols for animal care, magnetic resonance imaging, and anaesthesia were carried out under the authority of personal and project licenses in accordance with the UK Animals (Scientific Procedures) Act 1986 (ASPA).

Anaesthesia was induced using intramuscular injection of ketamine (10 mg/kg) either combined with xylazine (0.125–0.25 mg/kg) or with midazolam (0.1 mg/kg) and buprenorphine (0.01 mg/kg). Macaques also received injections of atropine (0.05 mg/kg intramuscularly), meloxicam (0.2 mg/kg intravenously) and ranitidine (0.05 mg/kg intravenously). Anaesthesia was maintained with isoflurane. The anaesthetized animals were either placed in an MRI compatible stereotactic frame (Crist Instrument Co., Hagerstown, MA) or resting on a custom-made mouth mold (Rogue Research, Mtl, QC, CA). All animals were then brought in a horizontal 3T MRI scanner with a full-size bore. Resting-state fMRI data collection commenced approximately 4 hr after anaesthesia induction, when the peak effect of ketamine was unlikely to be still present. In accordance with veterinary instruction, anaesthesia was maintained using the lowest possible concentration of isoflurane gas. The depth of anaesthesia was assessed using physiological parameters (continuous monitoring of heart rate and blood pressure as well as clinical checks for muscle relaxation prior to scanning). During the acquisition of the MRI data, the median expired isoflurane concentration was 1.083% (ranging between 0.6% and 1.317%). Isoflurane was selected for the scans as resting-state networks have previously been demonstrated to closely match known anatomical circuits using this agent (*Neubert et al., 2014*; *Vincent et al., 2007*). Slight individual differences in physiology cause slight differences in the amount of anaesthetic gas concentrations needed to impose a similar level of anaesthesia on different monkeys.

All but one animal were maintained with intermittent positive pressure ventilation to ensure a constant respiration rate during the functional scan; one macaque was breathing without assistance. Respiration rate, inspired and expired $CO_2$, and inspired and expired isoflurane concentration were monitored and recorded using VitalMonitor software (Vetronic Services Ltd., Devon). In addition to these parameters, core temperature was monitored using a Opsens temperature sensor (Opsens, Quebec, Canada) and pulse rate, $SpO_2$ (>95%) were monitored using a Nonin sensor (Nonin Medical Inc, Minnesota) throughout the scan.

A four-channel phased-array radio-frequency coil in conjunction with a local transmission coil was used for data acquisition (Dr. H. Kolster, Windmiller Kolster Scientific, Fresno, CA). Whole-brain blood oxygen level dependent (BOLD) fMRI data were collected for 1600 volumes from each animal (except for one with 950 volumes), using the following parameters: 36 axial slices, in-plane resolution 1.5 × 1.5 mm, slice thickness 1.5 mm, no slice gap, TR = 2280 ms, TE = 30 ms. Structural scans with a 0.5 mm isotropic resolution were acquired for each macaque in the same session, using a T1-weighted MP-RAGE sequence.

## Human

Twenty volumetric (as opposed to grayordinate) resting state fMRI datasets (Age 22–35 years; 13 male) were downloaded from the Human Connectome Project (HCP) (UgurbilWU-Minn K, for the *Van Essen et al., 2013*). Whole-brain BOLD EPI images were collected for approximately 15mins (1200 volumes) using a standardised protocol (2 mm isotropic resolution, 72 slices, TR = 720

ms, TE = 33.1 ms, multiband factor = 8). Only the first session with phase encoding left-to-right (LR) was used. Rs-fMRI data were already pre-processed using the FIX pipeline (automatic ICA rejection and regression of 24 head motion parameters) (*Griffanti et al., 2014*; *Salimi-Khorshidi et al., 2014*) and normalised into MNI space.

## Resting state pre-processing and analysis

In all species, we used an ICA nuisance regression pre-processing strategy (FIX for human and mouse, manual component rejection for macaque). For all species, resting state analyses were conducted using CONN (*Whitfield-Gabrieli and Nieto-Castanon, 2012*). Data were bandpass filtered according to recent recommendations (Mouse: 0.01–0.25 Hz; Macaque: 0.01–0.087 Hz; Human: 0.01–0.15 Hz), linear detrended, and despiked. Lowpass filter limits were set to be 5 volumes of data in all species; however, this was further reduced for the human data in order to avoid physiological artefacts which are believed to occur at frequencies > 0.2 Hz (*Baria et al., 2011*).

Resting state analyses began with the creation of three template connectivity fingerprints in mice (CP.m, NAcc, CP.l) and macaques (caudate, NAcc, and putamen). For each dataset, we extracted the principle eigenvariate from three striatal seeds and correlated these with timeseries extracted from target regions. To confirm that of each striatal seed has a unique connectivity fingerprint we used the MrCat toolbox (https://github.com/neuroecology/MrCat) to establish the Manhattan distance between striatal connectivity fingerprints within species (i.e. comparing CP.m and CP.l fingerprints in mice) and used permutation testing (10,000 permutations) to test for significance. This same procedure was used to confirm that macaque striatal connectivity fingerprints were significantly different from one another.

Individual connectivity fingerprints within species were averaged (robust mean) to create template connectivity fingerprints for each striatal seed (see *Figures 2*, *4* and *6*). In the comparison species, we extracted a connectivity fingerprint for each striatal voxel (correlation between the voxel and target timeseries). This voxel-based fingerprint was correlated with each of the template fingerprints from a different species and the resulting correlation value assigned to the voxel. This produced three correlation maps for each participant (one for each striatal seed) describing the correlation between each voxel fingerprint and the template fingerprint. Maps were Fisher's r-to-Z transformed and run through permutation testing in FSL's randomise (10,000 permutations, TFCE corrected p<0.05) to establish which voxels showed a significantly similar connectivity fingerprint.

T-Maps generated in the previous step were used as ROIs in CONN to generate whole-brain connectivity maps. A weighted timeseries from only positive (i.e. similar) voxels was extracted and used to identify connected regions across all grey matter voxels within the right hemisphere. Significant connectivity was established using permutation testing (10,000 permutations) and correction for multiple comparisons (p<0.001 voxel threshold, cluster-extent p<0.05 FDR). A conjunction analysis was employed to compare connectivity maps for assigned and unassigned voxels (*Friston et al., 2005*; *Price and Friston, 1997*). This approach is a more stringent comparison as it requires connectivity in the unassigned voxel map to be significantly greater than the connectivity for all the assigned voxel maps (i.e. unassigned >CP.m and unassigned >NAcc and unassigned >CP.l).

## Anatomical and functional localisation

Mouse-to-human and macaque-to-human striatal homologs were localised using the Harvard Oxford subcortical atlas and a task-based parcellation of the striatum (*Pauli et al., 2016*). Rather than using the traditional approach of localisation based on local maxima, we used the distribution-based cluster assignment method outlined in *Eickhoff et al. (2007)* to highlight the central tendency of activations and avoid localising to peripheral structures. This approach compares probability distributions for the underlying anatomical regions with probability distributions for the functional activation cluster, allowing one to make judgments about whether brain regions are over or under-represented. Specifically, the mean probability for area X at the location of the functional activation is divided by the overall mean probability for area X in all voxels where it was observed. This provides a quotient which indicates how much more (or less) likely an area was observed in the functionally defined volume than could be expected if the probabilities at that location would follow their overall distribution. A quotient >1 indicates a rather central location of the activation with respect to this area, whereas a quotient <1 a more peripheral one. Cortical activations were localised using a

combination of cytoarchitectonic probability maps from the Anatomy Toolbox (*Eickhoff et al., 2007*; *Eickhoff et al., 2006*; *Eickhoff et al., 2005*) and connectivity-based parcellation maps available in FSLEYES (*Mars et al., 2013*; *Mars et al., 2011*; *Neubert et al., 2015*; *Neubert et al., 2014*; *Sallet et al., 2013*; *Tziortzi et al., 2014*). Cerebellar activations were localised using the probabilistic cerebellar atlas (*Diedrichsen et al., 2009*).

## Acknowledgements

VZ is supported by an SNSF AMBIZIONE grant PZ00P3_173984/1. JS is supported by a Wellcome Trust grant 105651/Z/14/Z. NW is supported by ETH Research Grant ETH-38 16–2. RBM is supported by the Biotechnology and Biological Sciences Research Council (BBSRC) UK [BB/N019814/1] and the Netherlands Organization for Scientific Research NWO [452-13-015]. The Wellcome Centre for Integrative Neuroimaging is supported by core funding from the Wellcome Trust [203139/Z/16/Z].

## Additional information

### Funding

| Funder | Grant reference number | Author |
| --- | --- | --- |
| Schweizerischer Nationalfonds zur Förderung der Wissenschaftlichen Forschung | PZ00P3_173984/1 | Valerio Zerbi |
| Wellcome | 105651/Z/14/Z | Jerome Sallet |
| Eidgenössische Technische Hochschule Zürich | ETH-38 16-2 | Nicole Wenderoth |
| Biotechnology and Biological Sciences Research Council | BB/N019814/1 | Rogier B Mars |
| Nederlandse Organisatie voor Wetenschappelijk Onderzoek | 452-13-015 | Rogier B Mars |
| Wellcome | 203139/Z/16/Z | Jerome Sallet Rogier B Mars |

The funders had no role in study design, data collection and interpretation, or the decision to submit the work for publication.

### Author contributions

Joshua Henk Balsters, Conceptualization, Data curation, Software, Formal analysis, Investigation, Visualization, Methodology, Project administration; Valerio Zerbi, Conceptualization, Resources, Data curation, Formal analysis, Funding acquisition, Investigation, Visualization, Methodology, Project administration; Jerome Sallet, Resources, Data curation, Supervision, Methodology; Nicole Wenderoth, Conceptualization, Supervision, Funding acquisition, Methodology; Rogier B Mars, Conceptualization, Software, Funding acquisition, Methodology, Project administration

### Author ORCIDs

Joshua Henk Balsters ![ORCID] https://orcid.org/0000-0001-9856-6990
Valerio Zerbi ![ORCID] https://orcid.org/0000-0001-7984-9565
Jerome Sallet ![ORCID] http://orcid.org/0000-0002-7878-0209
Nicole Wenderoth ![ORCID] http://orcid.org/0000-0002-3246-9386
Rogier B Mars ![ORCID] http://orcid.org/0000-0001-6302-8631

### Ethics

Human subjects: Human subjects: Fully described in the core HCP literature referenced here; the paper is only using publicly available datasets on human imaging data.
Animal experimentation: Mouse: Protocols for animal care, magnetic resonance imaging, and anesthesia were carried out under the authority of personal and project licenses in accordance with the

Swiss federal guidelines for the use of animals in research, and under licensing from the Zürich Cantonal veterinary office (license number ZH198/16). Macaque: All non-human primate procedures were carried out in accordance with Home Office (UK) Regulations and European Union guidelines (EU directive 86/609/EEC; EU Directive 2010/63/EU).

## Decision letter and Author response

Decision letter https://doi.org/10.7554/eLife.53680.sa1
Author response https://doi.org/10.7554/eLife.53680.sa2

## Additional files

### Supplementary files

• Supplementary file 1. Supplementary tables.

• Transparent reporting form

### Data availability

The human resting state fMRI data was obtained from the Human Connectome Project (www.humanconnectome.org).

The following datasets were generated:

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
