## [Decision Letter]

**Acceptance summary:**

This study uses a connectivity fingerprint matching approach to compare cortico-striatal connectivity profiles across species (mice, nonhuman primates, and humans). The findings show that connectivity patterns of the nucleus accumbens and cortico-striatal motor circuits are similar across all species, whereas those of the caudate and anterior putamen are unique for non-human primates and humans. Much clinical research is directly based on preclinical findings in mice, implicitly assuming correspondence across species. This is one of the rare studies that attempts to empirically test this assumption.

**Decision letter after peer review:**

Thank you for submitting your article "Primate homologs of mouse cortico-striatal circuits" for consideration by *eLife*. Your article has been reviewed by three peer reviewers, one of whom is a member of our Board of Reviewing Editors, and the evaluation has been overseen by Kate Wassum as the Senior Editor. The following individuals involved in review of your submission have agreed to reveal their identity: Mark Laubach co-reviewed with Samantha White (Reviewer #2).

The reviewers have discussed the reviews with one another and the Reviewing Editor has drafted this decision to help you prepare a revised submission.

Summary:

Using connectivity fingerprint matching, this study compares cortico-striatal connectivity profiles across species (mice, nonhuman primates, and humans). The results demonstrate that the connectivity patterns of the nucleus accumbens and motor control regions are similar across species, and thus conserved. In contrast, the cortico-caudate and anterior putamen connectively profiles are not similar.

All reviewers agreed that this is an interesting and important study. However, reviewers also raised a number of concerns regarding the interpretation and discussion of the findings which need to be addressed. The most important issue that was discussed among reviewers was whether a 3-fold parcellation is sufficient to capture the complexity of the striatum and whether more fine-grained parcellations would provide more valuable information. The authors are strongly encouraged to consider more fine-grained parellations.

Essential revisions:

1) How was the Allen Institute database of tracer experiments transformed to Figure 1—figure supplement 1? Based on the literature and the tracer injections in the Allen Institute database, there is more precision to cortico-striatal projections. The authors likely combined all the tracer injections located in each of the three striatal segments. However, the rational for parcellating the striatum into these three rather large regions is unclear. Based on anatomic maps across species, three regions are not likely to be sufficient, given the complexity of cortico-striatal projections (even in the rodent brain). If the authors want to establish an imaging fingerprint for the mouse, it might better to use the data from the Allan Institute to first develop a segmentation of the mouse striatum that contained more than three large areas. For example, the shell might be separated from the core. The ICP is quite large and should be further segmented into components based on cortical inputs. It would be important to segment the mouse striatum in more than 3 general areas.

2) Discussion was lacking on limitations of using resting state data to infer connectivity, especially in comparison to DTI (which your group has also used) as well as traditional tract-tracing methods (Heilbronner) and viruses (Ährlund-Richter). The manuscript would benefit from some additional text on these issues.

a) Why did you use resting state data and not DTI? Please add a comment in the manuscript.

b) Are there are any specific connections that you found that are not reported in traditional tracer studies? If so, how can you account for such data? Network effects? By contrast, are there known connections that you did not find any evidence for? Please add a comment about these issues in the manuscript.

c) Neuroimaging does not allow for examining cell specific connectivity, as in viral tracer studies, and it would be good if you acknowledged the recent study by Ährlund-Richter and colleagues (PMID: 30886408) and described the limits on your interpretations that are due to using imaging, and not cell specific tracing. For example, some effects in your functional connectivity analysis might be mediated by connections between interneurons and projection cells or by inputs to interneurons, and this would not be detectable with either your method or traditional tracers.

d) Related to this, several of the connections identified in the paper are somewhat surprising, given the known anatomy, i.e. connectivity of V1 to the nucleus accumbens. While imaging results don't necessarily reflect all direct connections, for the most part they should be somewhat related. A more thorough review of how the results interface with the known anatomic connections is important, especially in the case in which connections seem to outliers.

3) Throughout the paper, the term "rodent" is used to discuss anatomical data from studies in mice. You should revise your text to replace rodent with mice. In addition, your manuscript would benefit from some text discussing why you are using data from mice, and not from rats. Obviously, this has to do with the Allen atlas. But what are the limitations of basing understanding the rodent brain on this one organism? After all, they are separated by ~27M of divergent evolution (see Cao et al. (2000), PMID: 11163972).

4) In the text referring to Figure 1 (subsection “Mouse to human comparison”, first paragraph), you cite the study by Hunnicutt. It reported on anatomical parcellation, not functional data. It would be good to cite primary functional studies at this point in the manuscript. However, here you must be careful to only cite studies in mice, unless you are also willing to explain why you would use studies in rats to explain function and studies in mice to explain structure throughout the manuscript). Maybe best would be to cite what you can for evidence for effects of lesions etc on learning, flexibility, decision making etc in mice, and also follow up with citations from the larger behavioral literature on rats. And then comment on the need for more large scale anatomy in rats and more carefully done operant studies in mice.

5) The conclusions that the connectivity patterns for the nucleus accumbens and motor areas across species, but not the caudate nucleus and anterior putamen is to be expected, and not surprising. The conservation of connectivity profiles across species for the n. accumbens and motor results are consistent with the well-known connections of these two evolutionarily conserved areas that mediate basic functions- appetitive and motor behaviors. The evolutionary expansion of cortex is primarily in cognitive areas, which are directly linked to the caudate and anterior putamen. This should be discussed.

6) Too little is said about whether or not the target brain areas that constitute the fingerprint can be considered homologous across the three species. The current analysis is just as much a test of the similarities in the striatum as it is a test of whether or not the target areas are similar across species. The degree to which the latter is true will affect the results and conclusions about striatal circuits. I think this deserves a bit more attention in the Discussion.

---

## [Author Response]

Essential revisions:1) How was the Allen Institute database of tracer experiments transformed to Figure 1—figure supplement 1? Based on the literature and the tracer injections in the Allen Institute database, there is more precision to cortico-striatal projections. The authors likely combined all the tracer injections located in each of the three striatal segments. However, the rational for parcellating the striatum into these three rather large regions is unclear. Based on anatomic maps across species, three regions are not likely to be sufficient, given the complexity of cortico-striatal projections (even in the rodent brain). If the authors want to establish an imaging fingerprint for the mouse, it might better to use the data from the Allan Institute to first develop a segmentation of the mouse striatum that contained more than three large areas. For example, the shell might be separated from the core. The ICP is quite large and should be further segmented into components based on cortical inputs. It would be important to segment the mouse striatum in more than 3 general areas.

We’d like to thank the reviewers for opportunity to address these points. Regarding the first point, “How was the Allen Institute database of tracer experiments transformed to Figure 1—figure supplement 1?”. We have added the following information to the Materials and methods section ‘Mouse Striatal Seeds’.

“Allen Institute database of tracer experiments were obtained and processed as described in Grandjean et al., 2017. [...] The connectivity between striatal voxels and cortical seeds was determined after normalizing the fluorescence with the volume of injection (for further details, see Oh et al., 2014).”

Regarding the second point “the rational for parcellating the striatum into these three rather large regions is unclear”. Figure 1 now provides a more detailed schematic explaining how we used tracer label to create mouse striatal seeds. Each striatal voxel had 68 values (one per tracer injection site) indicating the density of tracer label from each injection site terminating at that voxel. As such, each voxel possessed a connectivity fingerprint dictating the pattern of cortico-striatal tracer-based connectivity for each injection site. The three segments were derived through a data-driven approach using hierarchical clustering to sort a correlation matrix of values indicating how similar the tracer-based connectivity fingerprint of each voxel was to one another, i.e. does a neighbouring voxel have the same connectivity pattern or a different one? Finding the optimal solution using such a method is not a simple subject (see Eickhoff et al., 2015). Previously, we have used silhouette value as a way to find the solution that maximises within-segment similarity and between-segment differences. Using this approach, the optimal solution was a three segment solution. Figure 1 includes the silhouette values for each solution (2-20 segments) highlighting that the 3 segment solution was the optimal solution.

To further validate this three segment solution, we investigated cortico-striatal connectivity in our rsfMRI data using an independent 33 segment parcellation of the striatum recently published by Chon et al., 2019. However, the parcellation from Chon et al., 2019, was derived from the cortico-striatal projectome at a much finer spatial resolution than our rsfMRI data. When using the same data-driven approach as applied to the tracer-based data, our hierarchical clustering analysis revealed that the optimal solution was 4 clusters of segments; NAcc, CP.l, CP.m, and a section of the CP.tail. This indicates a high degree of overlap between the 33 cortico-striatal connectivity fingerprints, most likely because the nuanced connectivity differences that distinguished each of the 33 striatal segments in the original study were not visible at the coarser resolution of our rsfMRI dataset. We are not suggesting here that the Chon parcellation is wrong, rather that these subtle differences between segments were not visible at the resolution of our data. Note that the clustering results of the Chon parcellation showed a remarkable consistency with the results of the voxelwise tracer-based analysis (quantified using dice similarity – see Figure 1—figure supplement 1). Here, the CP.tail fails to have a corresponding match (all spatial correlations are 0). This is because the CP.tail segment falls entirely outside of the striatal mask used in the previous analysis and included voxels in the bordering lateral amygdala nucleus. Given spatial smoothing involved in fMRI analysis it is commonplace to remove voxels that border adjacent structures to avoid signal contamination which is known to have a significant effect on network detection (Smith et al., 2011). We suggest that this CP.tail segment is likely to be distinct because of this signal contamination which is additionally problematic as the basolateral amygdala is one of our targets. As such we have kept the same translational analysis across species based on three cortico-striatal segments defined using an independent tracer dataset and included this additional analysis in Figure 1—figure supplement 1.

Given the consistency between both tracer-based and rsfMRI-based segmentations of the mouse striatum, we believe that at this resolution a coarse three segment description is optimal. However, we would agree with the reviewers that whilst such a solution is optimal given the resolution of the data, it is unlikely to fully capture the complexity of cortico-striatal circuits. As such, we have also extended our previous discussion of this caveat to include additional discussion of the recent atlas by Chon et al., 2019, and examples from studies of the human cortico-striatal system such as the recent work by Tian et al., 2020. A copy of this new section is included below,

“One caveat to the approach used here is the definition of striatal seeds in the mouse. […] We note, however, that the current data are of the best resolution and quality currently in general use (Grandjean et al., 2020; Marcus et al., 2016; Milham et al., 2018Click or tap here to enter text. and it is likely most translational studies will not be working with data of superior quality.”

2) Discussion was lacking on limitations of using resting state data to infer connectivity, especially in comparison to DTI (which your group has also used) as well as traditional tract-tracing methods (Heilbronner) and viruses (Ährlund-Richter). The manuscript would benefit from some additional text on these issues.

We agree that the following issues require further discussion. We have now amended our ‘Conclusions and cautions’ section of the Discussion to create a larger methodological caveats section that addresses all of the issues raised in these points. We have copied the entire section below for the authors to read, and dealt with each section (a-d) individually within this document.

“Caveats and cautions

Here, we used rsfMRI as a common methodology to compare connectivity fingerprints across humans, nonhuman primates, and mice. […] As this line of research continues, it will be possible to refine our target models based on the results of previous studies.”

a) Why did you use resting state data and not DTI? Please add a comment in the manuscript.

In our previous work, we used DTI to infer anatomical similarities between macaques and humans (e.g., Mars et al., 2016; 2018; reviewed in Thiebaut de Schotten et al., 2018), but also resting state fMRI (Mars et al., 2011; Neubert et al., 2014; 2015). For this work we focused on mouse-human and mouse-macaque similarities and we decided resting-state fMRI would be a better connectivity metric. Our prior work demonstrated the strong agreement between inter-modal functional connectivity obtained through rsfMRI (i.e. functional connectome) and the anatomical ground truth provided by viral tracer injections in the mouse brain (i.e. structural connectome) (Grandjean et al., 2017). Specifically, for cortico-striatal connectivity, we found that approximately 88% of the voxels in the striatum have similar fingerprints between the two modalities. This provided the critical foundation for using rsfMRI connectivity as a translational tool to study inter-species brain connectivity similarities.

Whilst it is possible to obtain connectivity data in the mouse using diffusion-MRI data, previous studies have found a much poorer overlap between diffusion metrics and tracer data compared to rsfMRI. Calbrese et al., 2015, used ex-vivo samples and an extremely high resolution (43um isotropic), which required a scan time of 235 hours per sample. Despite the high quality of the diffusion tractography data, the authors observed relatively poor correspondence with neuronal tracer data, particularly with regard to 3D colocalization analysis. Strong correlation results were only observed in the coarsest, parent structure-level connectivity analysis, but were low for cortico-striatal connections. Similarly, another study (Sinke et al., 2018) confirmed that the degree of similarity between tractography-based connectivity and neuronal tracer data in rats is low and led to a substantial number of false positive and false negative connections. This issue was apparent even when using the most sophisticated tractography algorithms, including multi-shell multi-tissue constrained spherical deconvolution and global tractography. For these reasons, we decided not to include structural connectivity comparisons between species in the manuscript. We have included the following in the Discussion.

“DWI is another technique that can be used across species as an alternative to rsfMRI. […] In comparison, Grandjean et al., 2017, found 87-88% overlap between rsfMRI connectivity and tracer connectivity in cortico-cortico circuits and cortico-striatal circuits.”

b) Are there are any specific connections that you found that are not reported in traditional tracer studies? If so, how can you account for such data? Network effects? By contrast, are there known connections that you did not find any evidence for? Please add a comment about these issues in the manuscript.

Here, the reviewers have raised an important issue regarding the similarities and differences between methods of detecting brain connectivity. As the reviewers rightly point out, connectivity derived from rsfMRI is not limited to mono-synaptic connections and reflects whole network dynamics. That said, in our previous work (Grandjean et al., 2017), we showed that in mice there was 87-88% overlap between tracer-based connectivity and rsfMRI connectivity for the isocortex and striatum respectively. However, we also found that cortico-thalamic circuits failed to show a good overlap between methods (8.8%). We proposed that this could reflect the regional effects of anaesthesia in rsfMRI or the relationship between connectivity strength and the amount of terminal label used for each injection site in the Allen Institute tracer data. These issues were discussed in Grandjean et al., 2017, as this was the focus of that paper, whereas here our aim was to bypass differences in methods used across species and compare like-with-like (i.e. rsfMRI in all species). We have included these issues in our new Discussion section ‘Caveats and cautions’, but we believe an exhaustive comparison of tracer-based vs. resting state connectivity goes beyond the scope of this paper.

“The strong agreement between the rsfMRI and tracer modalities is perhaps surprising, since rsfMRI connectivity and studies on neuronal tracers are fundamentally different. […] As such, it is crucially important to compare like-with-like using the same methods across species.”

c) Neuroimaging does not allow for examining cell specific connectivity, as in viral tracer studies, and it would be good if you acknowledged the recent study by Ährlund-Richter and colleagues (PMID: 30886408) and described the limits on your interpretations that are due to using imaging, and not cell specific tracing. For example, some effects in your functional connectivity analysis might be mediated by connections between interneurons and projection cells or by inputs to interneurons, and this would not be detectable with either your method or traditional tracers.

We agree with the reviewers that MRI is clearly limited in cellular specificity, as highlighted by the exquisite work of Ährlund-Richter et al., 2019. However, we argue in the Introduction that for all the specificity of tracer studies, such methods are rarely applied to humans making them less useful for comparative neuroscience. Here, we have aimed to harmonise connectivity research across humans, nonhuman primates, and mice by using the same (all-be-it less specific) methodology. We have added these points into the ‘Caveats and cautions’ section:

“Tracer based methods can also provide details regarding cell specific connectivity that distinguishes between different classes of interneurons. […] While rsfMRI is unlikely to ever reach the level of specificity achieved by tracer methods, we believe the ability to use this technique in all species makes it a powerful tool for comparative neuroanatomy.”

d) Related to this, several of the connections identified in the paper are somewhat surprising, given the known anatomy, i.e. connectivity of V1 to the nucleus accumbens. While imaging results don't necessarily reflect all direct connections, for the most part they should be somewhat related. A more thorough review of how the results interface with the known anatomic connections is important, especially in the case in which connections seem to outliers.

As previously stated, we believe a thorough comparison of tracer-based connectivity and resting state connectivity is beyond the scope of this paper. In the majority of cases we would suggest that our activations are in keeping with both tracer studies and previous rsfMRI studies (Choi et al., 2012; Haber, 2016). For example, it is clear that connectivity with our unassigned voxels tracks the boundaries of the frontoparietal network as defined by Yeo et al., 2011. Even some of the more surprising findings do in fact coincide with known anatomy. For example, Hunnicutt et al., 2016, and Khibnik et al., 2014, show visual projections to the dorsomedial striatum in mice. In primates, this projection is likely to be conserved through the inferior longitudinal fasciculus which is known to enter the Muratoff bundle and terminate in the body of the caudate nucleus (Saint-Cyr et al., 1990; Schmahmann and Pandya, 2007). As such there is evidence in both mice and primates for anatomical connections between occipital lobe and the caudate nucleus to facilitate this rsfMRI activation. Unfortunately, it is not possible to say whether these results are due to monosynaptic or polysynaptic effects. This caution has been raised in the ‘Caveats and cautions’ section of the Discussion.

“The level of detail available using tracer methods goes far beyond what is currently possible using fMRI. […] It is unclear whether this reflects a polysynaptic network effect or whether this reflects a change in monosynaptic connections between the occipital lobe and basal ganglia previously shown in mice (Hunnicutt et al., 2016; Khibnik et al., 2014) and primates (Saine-Cyr et al., 1990; Schmahmann and Pandya, 2006).”

3) Throughout the paper, the term "rodent" is used to discuss anatomical data from studies in mice. You should revise your text to replace rodent with mice. In addition, your manuscript would benefit from some text discussing why you are using data from mice, and not from rats. Obviously, this has to do with the Allen atlas. But what are the limitations of basing understanding the rodent brain on this one organism? After all, they are separated by ~27M of divergent evolution (see Cao et al. (2000), PMID: 11163972).

We thank the reviewer for addressing this important point. Recent studies have shown reproducible and qualitatively similar resting-sate networks in both mouse and rat (Zhiwei et al., 2018; Grandjean et al., 2020). However, no one has quantitatively and systematically assessed the similarities and differences between resting-state fMRI connectivity in rats and mice. Thus, we agree with the reviewer that the term “rodent” was improperly used in a number of instances. To correct this, we have changed rodent to either ‘rat’ or ‘mouse’ when referring to a specific paper or dataset which uses a specific species. However, we have kept the term rodent in a small number of instances, for example, “there is a vital need to harmonise findings across species by establishing similarities and differences in rodent and primate neuroanatomy”. In this instance we believe the comment applies to both rats and mice and therefore warrants the use of the term rodent.

This increased specificity also led us to address the reviewers second point, why use mice instead of rats? According to Dietrich et al., 2014, and UK Home Office statistics from 2018 (https://assets.publishing.service.gov.uk/government/uploads/system/uploads/attachment_data/file/835935/annual-statistics-scientific-procedures-living-animals-2018.pdf) mice are the most used mammalian species in scientific research. In addition, there are a wealth of freely available neuroscientific resources in mice that are not available in rats including tracer-based connectivity data from the Allen Institute, as well as the new mouse rsfMRI database (Grandjean et al., 2020). We have included these points in the Introduction.

“Animal models are currently providing crucial insights into neural structure, function, and disorders. […] However, with this comes a growing necessity for translational, comparative neuroscience to harmonise these results with our understanding of structure, function, and disease in the human brain.”

4) In the text referring to Figure 1 (subsection “Mouse to human comparison”, first paragraph), you cite the study by Hunnicutt. It reported on anatomical parcellation, not functional data. It would be good to cite primary functional studies at this point in the manuscript. However, here you must be careful to only cite studies in mice, unless you are also willing to explain why you would use studies in rats to explain function and studies in mice to explain structure throughout the manuscript). Maybe best would be to cite what you can for evidence for effects of lesions etc on learning, flexibility, decision making etc in mice, and also follow up with citations from the larger behavioral literature on rats. And then comment on the need for more large scale anatomy in rats and more carefully done operant studies in mice.

We agree with the reviewer and have removed the reference to Hunnicutt here. As the reviewer points out most studies supporting three functional subdivisions of the striatum are based on primate and rat research. We have replaced the Hunnicutt reference with the following optogenetic mice studies investigating motor control (Freeze et al., 2013), reward processing (Gunaydin et al., 2014), and reinforcement learning (Kravitz et al., 2012). We have also added the following warning to the striatum section of the Discussion, highlighting the species differences.

“While many studies in mice refer to a three domain striatal model (i.e. associative, limbic, motor), this concept is largely based upon research in non-human primates and rats (Balleine et al., 2009; Thorn et al., 2010; Yin and Knowlton, 2006). Although some anatomical studies in mice are providing greater support for the three domain model (Hunnicutt et al., 2016) there is a need for more operant studies in mice and more large scale anatomical studies in rats.”

5) The conclusions that the connectivity patterns for the nucleus accumbens and motor areas across species, but not the caudate nucleus and anterior putamen is to be expected, and not surprising. The conservation of connectivity profiles across species for the n. accumbens and motor results are consistent with the well-known connections of these two evolutionarily conserved areas that mediate basic functions- appetitive and motor behaviors. The evolutionary expansion of cortex is primarily in cognitive areas, which are directly linked to the caudate and anterior putamen. This should be discussed.

We are pleased that the reviewers agree with our findings. We had previously focussed on evolution from an anatomical perspective, i.e. the evolution of PFC, and cerebellum. However, we believe it is also worth discussing the behavioural changes co-occurring with evolution. As such we have included the following section in the Discussion (striatum section).

“Rather, these results are likely to reflect evolutionary pressures acting upon brain circuits underpinning cognitive behaviours (Passingham and Wise, 2012. […] As such we would caution researchers using mouse models for disorders predominantly affecting these processes as there appears to be less clear translation between mice and humans.”

6) Too little is said about whether or not the target brain areas that constitute the fingerprint can be considered homologous across the three species. The current analysis is just as much a test of the similarities in the striatum as it is a test of whether or not the target areas are similar across species. The degree to which the latter is true will affect the results and conclusions about striatal circuits. I think this deserves a bit more attention in the Discussion.

We agree that a key feature of the translational analysis is the choice of targets. Indeed, this was an aspect of the study we spent a great deal of time on, and Supplementary tables 1-3 in Supplementary file 1 provide citations justifying each of our targets. In the case of human and macaque targets, we used targets that were justified in multiple previously published studies (Mars et al., 2011; 2013; Neubert et al., 2014; 2015; Sallet et al., 2013). The addition of mice forced us to identify which of the established targets was viable in a mouse model, leading us to create both a mouse (reduced) model with 12 targets and a macaque (extended) model with 17 targets. It’s worth noting that in our final analysis (macaque to human) we ran both the extended and reduced models and found that there was very little difference, supporting the idea that our core 12 target model was robust and unlikely to be influenced by 1-2 targets. We have added the following to our Discussion section on methods.

“As well as seed definition, connectivity fingerprint matching relies on the accurate definitions of target areas across species. […] As this line of research continues, it will be possible to refine our target models based on the results of previous studies.”